# Metallizing the Surface of Halloysite Nanotubes—A Review

**Abdul-Razak Masoud** [1] , **Femi Alakija** [1], **Mohammad Jabed Perves Bappy** [2], **Patrick A. S. Mills** [3]
**and David K. Mills** [1,2,3,*]

[1] Molecular Science and Nanotechnology Program, College of Applied & Natural Sciences,
Louisiana Tech University, Ruston, LA 71272, USA

[2] Nanosystems Engineering Program, College of Engineering and Sciences,
Louisiana Tech University, Ruston, LA 71272, USA

[3] School of Biological Sciences, Louisiana Tech University, Ruston, LA 71272, USA

[*] Correspondence: dkmills@latech.edu; Tel.: +318-257-2640; Fax: +318-257-4574

**Abstract:** Halloysite nanotubes (HNTs) have been shown to be ideal nanoparticles for polymer reinforcement, sustained drug release, nano-reactor synthesis, toxic material removal, regenerative medicine, and as a substrate material for nanostructured coatings. Metal and metal oxide nanoparticles have been used for centuries in various medical applications, primarily for their antimicrobial, antifungal, and antiviral properties. The focus of this review is the metallization of HNT surfaces. Different technologies use specific metal compounds and multi-step chemical reactions to metalize the HNT surface. This review begins with a brief overview of the current methods for metallizing the HNT surface. Our focus then provides a detailed study on specific applications of metal-coated HNTs (mHNTs) in the field of nanomedicine. The focus is on using mHNTs and Mhnt polymer composites in anti-infective therapy, immunotherapy, dentistry, regenerative medicine, and wound healing. The importance of HNTs in aerospace, defense, and industry has emerged, and the application potential and enormous market value for metal oxide nanoparticles is apparent. The commercialization potential of metal-coated HNTs is also discussed.

**Keywords:** coatings; halloysite; functionalization; metal nanoparticles; nanomedicine





## 1. Introduction

### 1.1. Structure and Properties of Halloysite

Halloysite nanotubes (HNTs) exist worldwide and are mined from various mineral deposits, making them an easily accessible nanomaterial [1]. These clay nanotubes are structured as a two-layered aluminosilicate, have predominantly hollow nanotubular structures in the submicron range, and are chemically identical to kaolin clay [1,2]. HNTs typically display an inner diameter between 15–50 nm and a length between 100–2000 nanometers (Figure 1). HNTs have a large surface area and can be loaded and coated with various materials, such as drugs, metals, and biomacromolecules [2,3]. A wide range of active agents, including antibiotics, cancer drugs, marine biocides, and biological molecules, can be entrapped within the inner lumen and void spaces within the aluminosilicate shells [3,4].

### 1.2. Surface Modification of Halloysite

Due to their tubular microstructure, biocompatibility, high mechanical strength, thermal stability, and potential for surface modification, HNTs have seen use as additives to inorganic/polymeric hybrid materials, including biofunctional composites, drug carriers, high-performance catalysts, nanoreactors, tissue engineering substrates, bone implants, and high-efficiency adsorbents [5–7] (Figure 2).

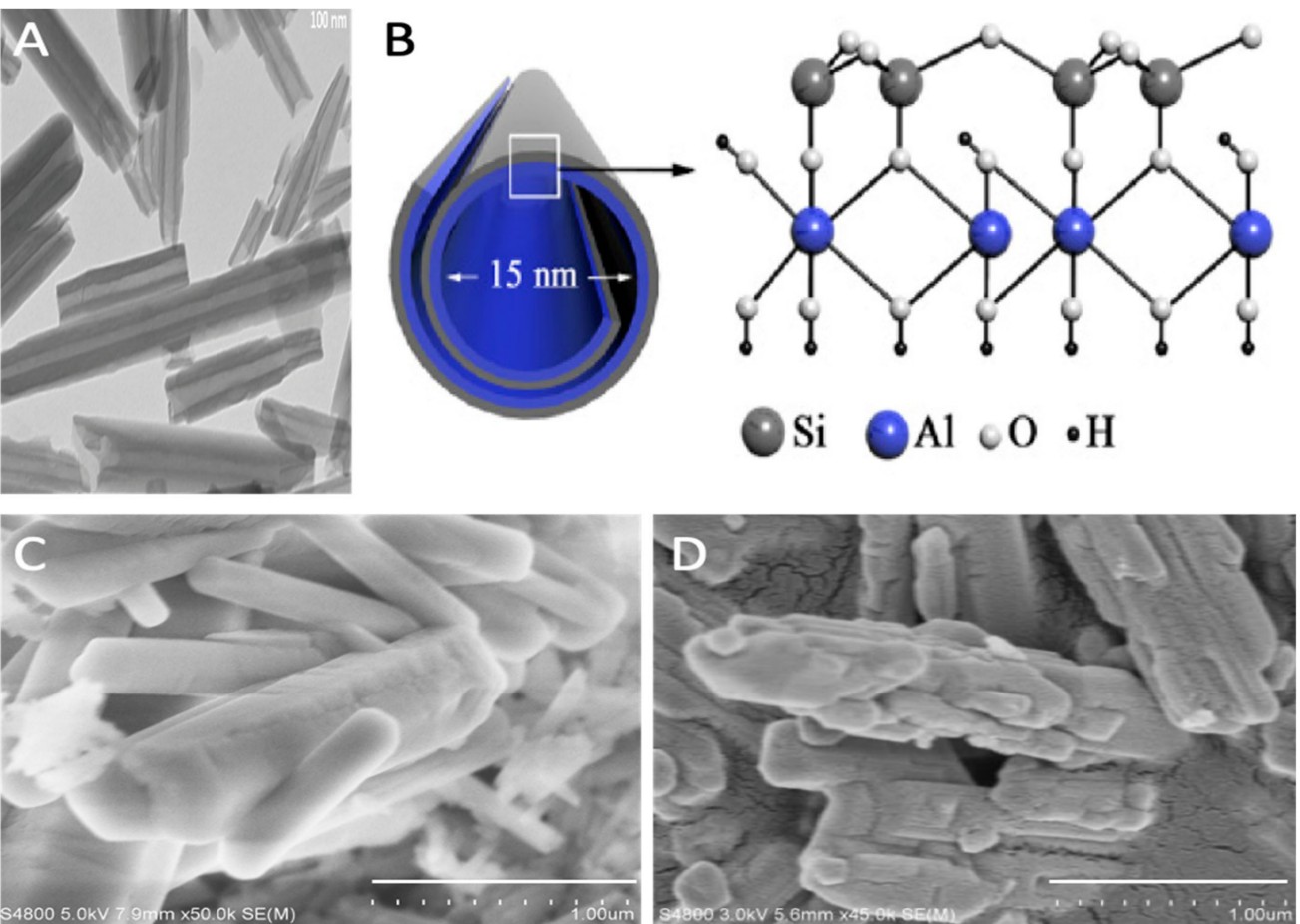

**Figure 1.** (**A**) TEM of halloysite showing the prominent lumen. (**B**) Chemical structure of halloysite. (**C**) Magnesium-coated halloysite nanotubes. (**D**) Strontium-coated halloysite nanotubes.

### 1.3. Halloysite Cyto- and Biocompatibility

Halloysite nanotubes are non-cytotoxic towards several cell types (up to 0.1 mg/Ml) including chondrocytes, dermal fibroblasts, osteoblasts, and stem cells on halloysite nanofilms or within HNT-hydrogel composites [6,8]. Examination of halloysite with in vitro assays shows that cells proliferated and maintained their cellular phenotype after HNT exposure [6]. Studies using *Caenorhabditis elegans* and *Paramecium caudatum* also found little evidence of cytotoxicity [9,10]. A recently completed study using several types of HNT tested them for cytotoxicity using human epithelial adenocarcinoma (HeLa), human hepatocellular carcinoma (HepG2), and Chinese hamster ovary (CHO) cell lines. The results demonstrated in vitro cytocompatibility at low doses and shorter exposure times, while higher amounts and more prolonged exposure demonstrated a significant decrease in cell vitality and an increase in genotoxic effects. This result indicates a clear need to exercise caution in using HNTs on living organisms.

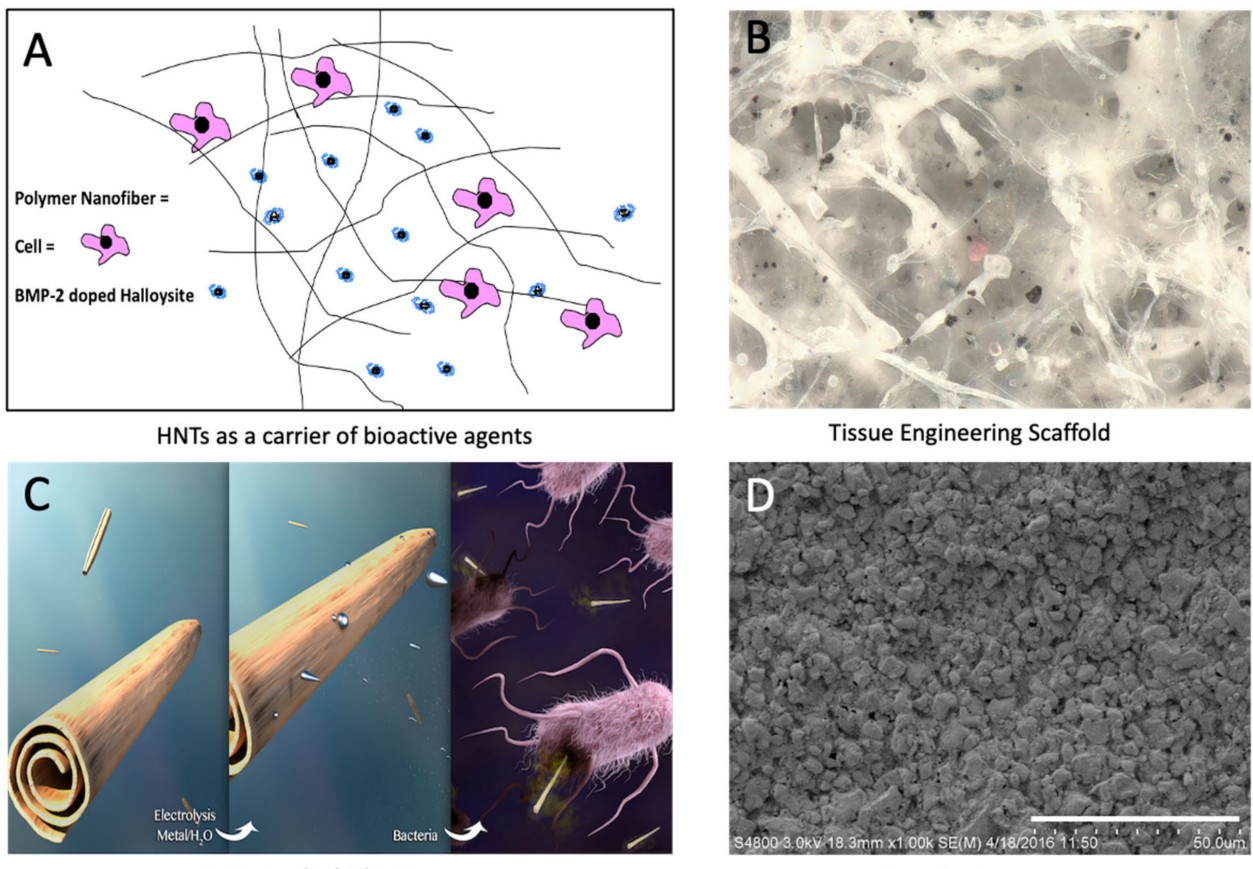

**Figure 2.** (**A**) Growth-factor-doped HNT nanofiber substrates. (**B**). Zinc-coated HNTs in PLA fibers for tissue regeneration. (**C**) Electrodeposition of silver nanoparticles onto the HNT surface for antimicrobial applications. (**D**) HNT and calcium phosphate cement for use as a bone implant.

There have only been a few reported results on HNT toxicity suggesting that they are a safe and ideal candidate for the development of new drug delivery systems and polymer-based tissue regeneration that can be widely used without severe side effects on living organisms. However, a complete halloysite biocompatibility study has not yet been carried out. Furthermore, studies with metal-oxide-coated HNTs must be conducted to assess their biocompatibility.

## 2. Metal Nanoparticles and Applications

Recent developments in metal nanoparticle synthesis and characterizations have led to several new biomedical technologies [11,12]. Current metal nanoparticle synthesis approaches include electrochemical [13], sonochemical [14], radiolytic [15], photochemical [16], sintering [17], and microbial synthesis [18] methods.

Several methods exist for the surface modification of halloysite nanotubes with metals. The current state of the art consists of various technologies that use specific metal compounds and multi-step chemical reactions. Most of the metal synthesis methods are multi-step chemical reactions to generate metallized nanoparticles that are costly, cumbersome, and often produce noxious reactants as a by-product of production [19–22]. As an illustration, the current method for depositing gold nanoparticles on HNT surfaces uses gold compounds and hydrogen tetrachloroaurate, potassium tetrachloroaurate, and sodium tetrachloroaurate dihydrate. It is applied using multi-step reactions [23]. Reducing agents such as sodium borohydride, aluminum borohydride, sodium triacetoxyborohydride, sodium cyanoborohydride, oxalic acid, sodium oxalate, potassium oxalate, formaldehyde,

hydrazine, or hydroquinone are also used in the metalization process [18–23]. Current metalization techniques are discussed in the following sections.

### 3. Surface Modification of Halloysite Nanotubes with Metals

#### 3.1. Calcination of Halloysite Nanotubes

Iron and nickel were deposited on halloysite through mixing metal compounds and calcination with high temperatures [22]. In addition, halloysite-supported cobalt was achieved through the calcination of halloysite and cobalt nitrite under temperatures as high as 623 Kelvin [20]. HNTs were also coated with metal acetylacetonates through a dry sintering process that did not use toxic chemicals, thus eliminating exposure to toxic chemicals and costs associated with chemical waste disposal [24].

#### 3.2. Metal Sintering of Halloysite Nanotubes

At 300 °C, most metal acetylacetonates break down entirely, releasing acetone and carbon dioxide while leaving positively charged metal ions behind [17,24]. These metal ions attach to the outside surface of halloysite due to its negatively charged outer lumen, allowing for a one-step sintering coating [17,24]. This coating method enables greater deposition of metal acetylacetonates onto the HNT surface as required for the desired application (Figure 3). Furthermore, chemical processes using toxic chemicals are not required, eliminating exposure to toxic chemicals and costs associated with the disposal of the resultant chemical waste [24].

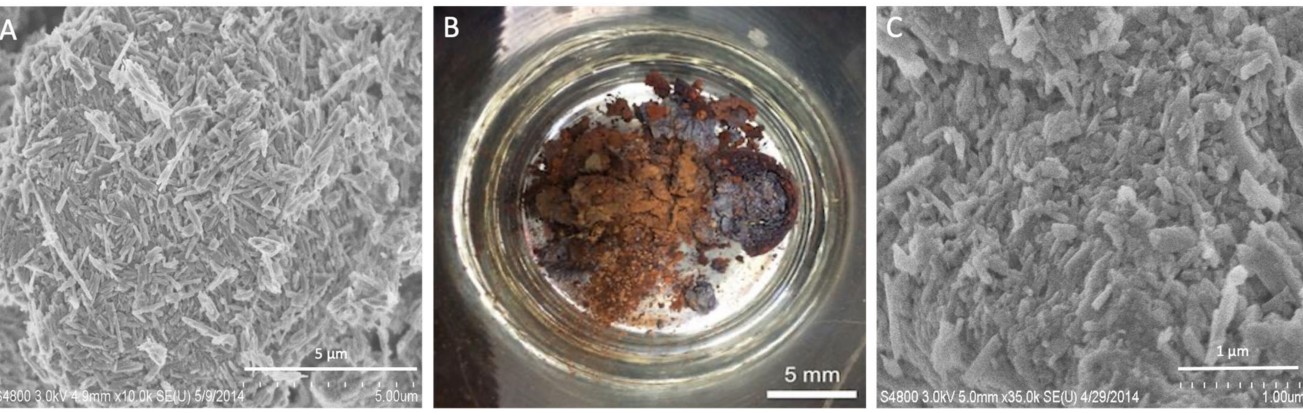

**Figure 3.** Iron-sintered halloysite. (**A**) Scanning electron micrograph (SEM) of native halloysite. (**B**). Iron-coated halloysite post-sintering. (**C**) SEM of sintered iron-coated halloysite.

#### 3.3. Electrodeposition for Metallizing HNTs

Generating and storing electricity has tremendous impact on technological advancements and human civilization. Henry Moissan demonstrated the potential use of electrolysis in industry and science when he isolated fluorine [25]. The phenomenon of electrolysis was a crucial chemical method that helped shape chemistry and manufacturing. The technique uses direct current (D/C) to drive an otherwise non-spontaneous chemical reaction through a solution. Electrolysis has been shown to have remarkable efficiency in the separation of materials. Several electrolysis methods have since been developed for industry, including electro-refining, electrosynthesis, the chloro-alkali process, the mercury cell process, the diaphragm cell process, and the membrane cell process. Electrolysis has fueled the development of many industries, including aluminum and sodium hydroxide production. This method continues to be a valuable technique for chemists, scientists, and industrial-scale output [26].

A series of studies examined metal oxides as a substrate for the deposition of various metal nanoparticles on halloysite using electrodeposition synthesis [27]. Metallized HNTs were monitored through scanning and transmission electron microscopy, X-ray diffraction,

and Fourier transform infrared spectroscopy. The results indicate the feasibility of electrodeposition as a new method for coating HNTs with metals (mHNTs) [28–30]. It is a green and straightforward method that is also an efficient and replicable nano-scale halloysite surface modification method. The cumulative data suggest that these inorganic hybrid nanocomposites may have potential applications in nano-bio sensors, biomedical devices, theragnostic systems, radiation adsorption composites, industrial catalytic systems, and antibacterial nanomaterials.

The advantages of this Invention over existing methods include the rapid one-step electrolysis fabrication technique and the associated low cost of using pure metals, pure water, and low-voltage for depositing MNP on HNT surfaces. Overall, this method allows HNT-MNP composites to be fabricated quickly with minimal starting materials.

### 3.4. Physical Adsorption or Self-Assembly

Clay minerals of the kaolin group, such as halloysite nanotubes, have been shown to have the potential to adsorb metal ions on their surfaces and lumens. Mellouk et al. demonstrated that this vital property is enhanced by initial intercalation with different functional groups, specifically $Na^+$, $NH_4^+$, or $Pb^{2+}$ acetate, prior to adsorbing metal ions onto the HNTs [31]. Positively charged copper (II) ions are attached to the negatively charged binding sites of halloysites through electrostatic interaction between the molecules as well as an ion-exchange mechanism between the copper (II) ions and the cations bound to acetate [31].

Another plausible mechanism involves the initial modification of the metal nanoparticles and subsequent immobilization on the surface of the HNTs. For example, $Fe_3O_4$ nanoparticles were modified with L-lysin and immobilized on the surface of HNTs via self-assembly through electrostatic interactions [32]. Alternatively, zinc ions were shown to be removable from aqueous solutions using HNTs without prior modifications through attachment to the halloysite surface and the inner lumen [33]. Silver ions were also adsorbed at increasing rates onto the surface of unmodified halloysite in aqueous solution with increasing Ph and temperature [34].

### 3.5. Solvothermal

Solvothermal synthesis of metal-HNT conjugates generally involves the modification of HNT nanotubes via a solvent-mediated reaction in a high-pressure-withstanding vessel at temperatures surpassing the boiling point of the reaction medium [35]. This method is advantageous because it has been streamlined into a one or two-step process with tuneable reaction parameters such as solute concentration, type of substrate, Ph, and pressure, allowing adequate control of particle distribution and size [36] or other factors [37].Titanium dioxide ($TiO_2$) [38] and zinc oxide (ZnO) [37] have commonly been deposited on the surfaces of halloysite nanotubes for varying applications. Wang et al. deposited TiO2 on halloysite nanotubes for improved photocatalytic activity and adsorption during the photocatalytic degradation of organic pollutants. The authors incubated halloysites in isopropanol under slightly acidic conditions while allowing adequate reaction time [38]. For the coating of halloysites with ZnO in the lumen and the outer surface, initial activation of both halloysite surfaces with NaOH was required. Zinc nitrate solute was deposited on a halloysite substrate under high temperature and pressure using an aqueous solvent [37]. Increased reaction times resulted in the accumulation of sufficient reactive ions on the halloysite substrate and their subsequent deposition onto the desired architecture. Manipulating temperature and pressure in the reaction vessel resulted in the optimal dissolution/diffusion of reactants and the attaining of a supercritical reaction state, respectively [35].

## 4. Biomedical Applications

Excessive and ill-advised use of antibiotics and self-medication have led to multiple forms of drug-resistant bacteria [39,40]. Various antimicrobial biomaterials, such as pro-

teins, enzymes, drugs, nanoparticles, metal salts, and metal ions, have been used with limited impact due to associated side effects, economic viability, and environmental sustainability. Many studies have reported that several strains of bacteria have developed resistance to β-lactamase activity, thus conferring immunity against the β-lactamase moiety of antibiotics [40,41]. In this regard, gold nanoparticles have shown antimicrobial effects against a variety of bacteria such as *E. coli* [42], *Streptococcus mutans* [43], and *Salmonella typhi* [44]. In addition, several research papers point out the enhanced synergistic activity of antibiotics when paired with metal nanoparticles [44].

Silver is a popular antimicrobial metal, and its various combinations and forms have been explored against a range of microorganisms. Still, unfortunately, many microbes have developed resistance against it over time [45]. In this context, gold is a candidate of choice similar to silver. When administered up to a threshold concentration, it has a minimal side effect on the human body [46].

Halloysite-nanotube-supported metal nanoparticles have potential uses in biomedical devices, antimicrobial surfaces, drug delivery systems, radiation-absorbent composites, plastic–elastomer composites, electronic components, and industrial catalysts (Figure 4). Metal nanoparticles (NPs), such as silver NPs, copper NPs, gold NPs, and others, have been deposited onto the surfaces of HNTs through various methods.

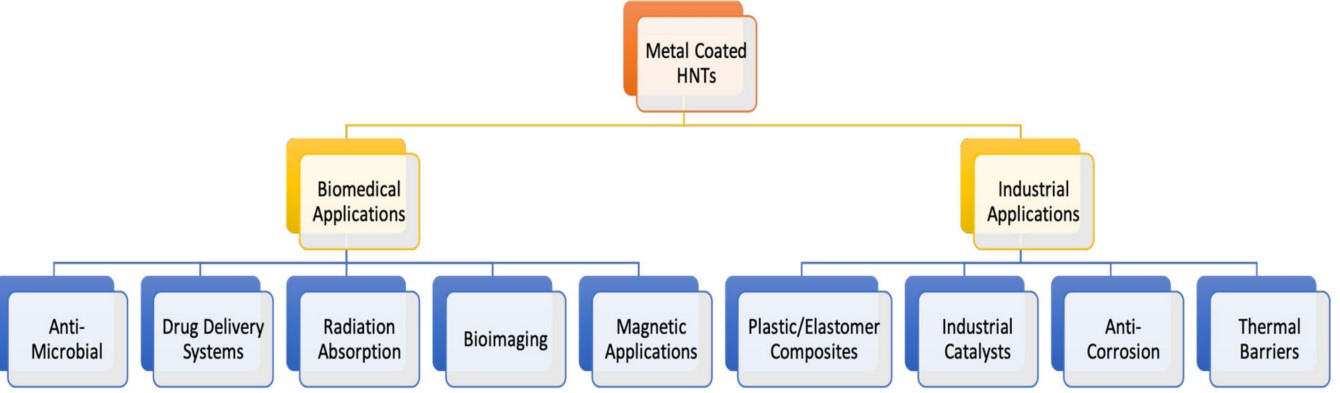

**Figure 4.** Application areas for metal-coated halloysite nanotubes [46].

*4.1. Antimicrobial Surfaces*

One major challenge for orthopedic surgeons in managing wound infection is how to manage the infections of implants and medical device surfaces. Any time there is a fracture, it is believed that it is contaminated. Therefore, surgeons have been making use of antibiotics for treatment and management. The current traditional standard of cure is microscopic-reservoir-type dressings that provide a barrier against external infections entering the body while promoting a moist wound healing environment; however, these do not change the topography of the wound itself. It has been shown that antibiotics reduce risk six-fold [47].

Over the years, antibiotics such as but not limited to tetracyclines, cephalosporins, quinolones, semi-synthetic penicillin, and non-antibiotics such as but not limited to chlorhexidine, nitrofurazone, and mupirocin have been incorporated into orthopedic devices as well as wound and fracture care products to prevent infections. However, limited success has been achieved. The solution to this problem that the medical community has been looking for is a topical antibacterial dressing that can provide an antimicrobial barrier for several days on a wound surface to reduce the frequency of painful dressing changes and, at the same time, provide a template to encourage cellular growth. Unfortunately, commercially available antimicrobial agents such as disinfectants and antibiotics are sometimes not very effective against new multiresistant pathogenic strains, and traditional means of disinfecting surfaces in hospitals such as spray, liquid, and powder are not able to provide prolonged sterile conditions [48].

Recently, surfaces have been coated with antimicrobial products to reduce microbial growth on the surface in several ways; however, the setbacks of these ways are shown by many healthcare provider reviews that indicate that a lack of sturdiness is a significant concern in antimicrobial products (Figure 5). Swaminathan et al. (2019) documented that the current way antimicrobial coating works on a surface is by altering its physicochemical properties or creating a type of coating that can provide a way to inhibit cell adhesion and biofilm formation [49]. The use of antimicrobial coating has started to attract interest in the hospital and transport fields. Lately, there has been an increase in public mobility by air, land (bus, car, train), and water (ships, boats). Many people touch many different parts of the transportation and hospital environments, eventually leading to the transfer of infections from one person to another. Therefore, the application of metal-based nanoparticles, which have been shown to have antimicrobial properties and sturdiness on surfaces, provides a potential means to eliminate the risk of infection.

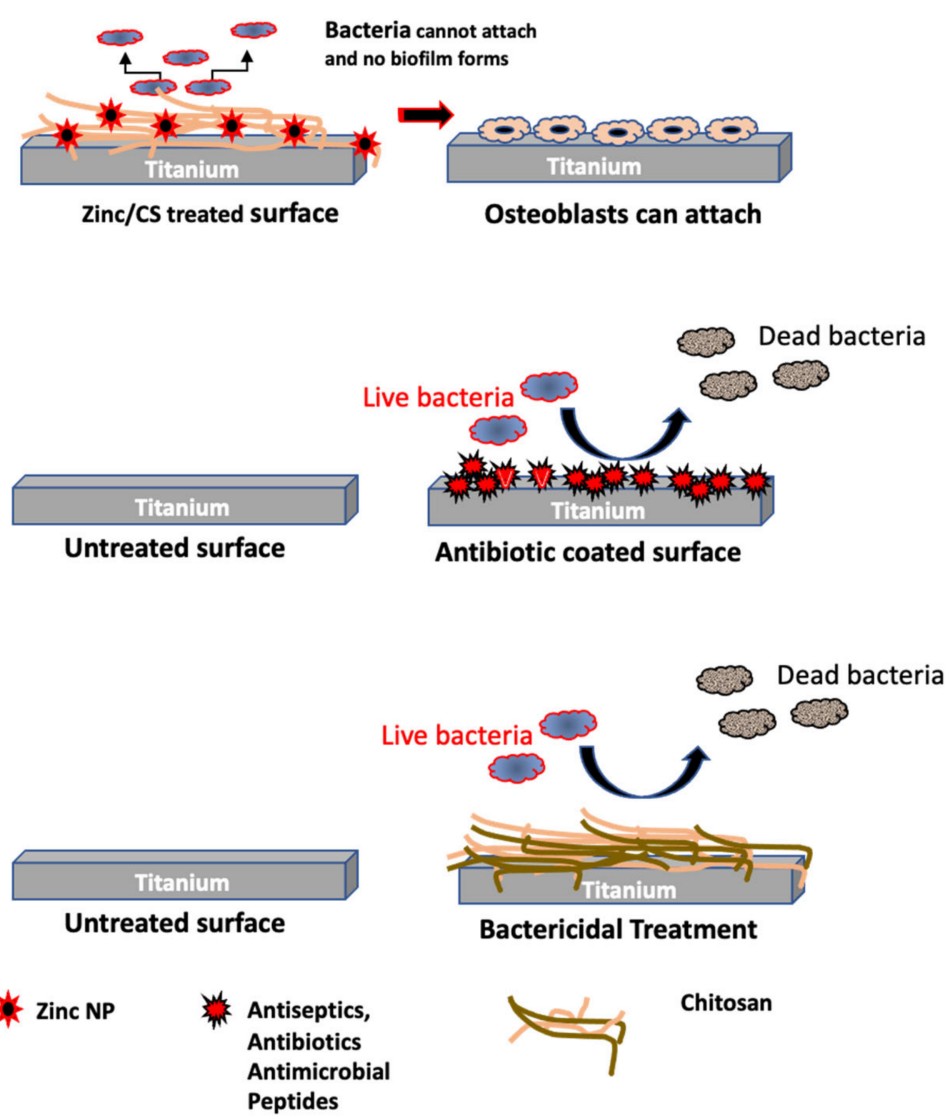

**Figure 5.** Surface modifications of surfaces can create bacteria-repellent surfaces.

Some metals and metal oxides such as Ag, Au, Cu, Pt, $Ag_2O$, $TiO_2$, $SiO_2$, CuO, ZnO, CaO, MgO, and $Fe_2O_3$ have been shown to have excellent antimicrobial activity against *E. coli*, B. subtilis, and other pathogenic microorganisms, which leads to their importance in preventing healthcare-associated infections that can be caused by frequently-touched metallic surfaces and medical devices in hospitals [49].

### 4.2. 3D Printed Biomedical Devices

Three-dimensional printing has demonstrated its usefulness in providing biomedical solutions to many clinical problems [50,51]. Complex geometries, sophisticated shapes, and designs, including unique internal and external architectures, are possible.

Three-dimensionally printed devices, combined with metal oxide nanoparticles, bioplastics (polycaprolactone, PCL, polylactic acid, PLA), resins (polymethylmethacrylate, PMMA), and halloysite nanotubes (HNTs) are under development for bone tissue regeneration and wound healing [52,53]. These devices provide sustained drug delivery that addresses wound site and bone infections by providing high concentrations of antibiotics to the affected site while enhancing tissue regeneration (Figure 6).

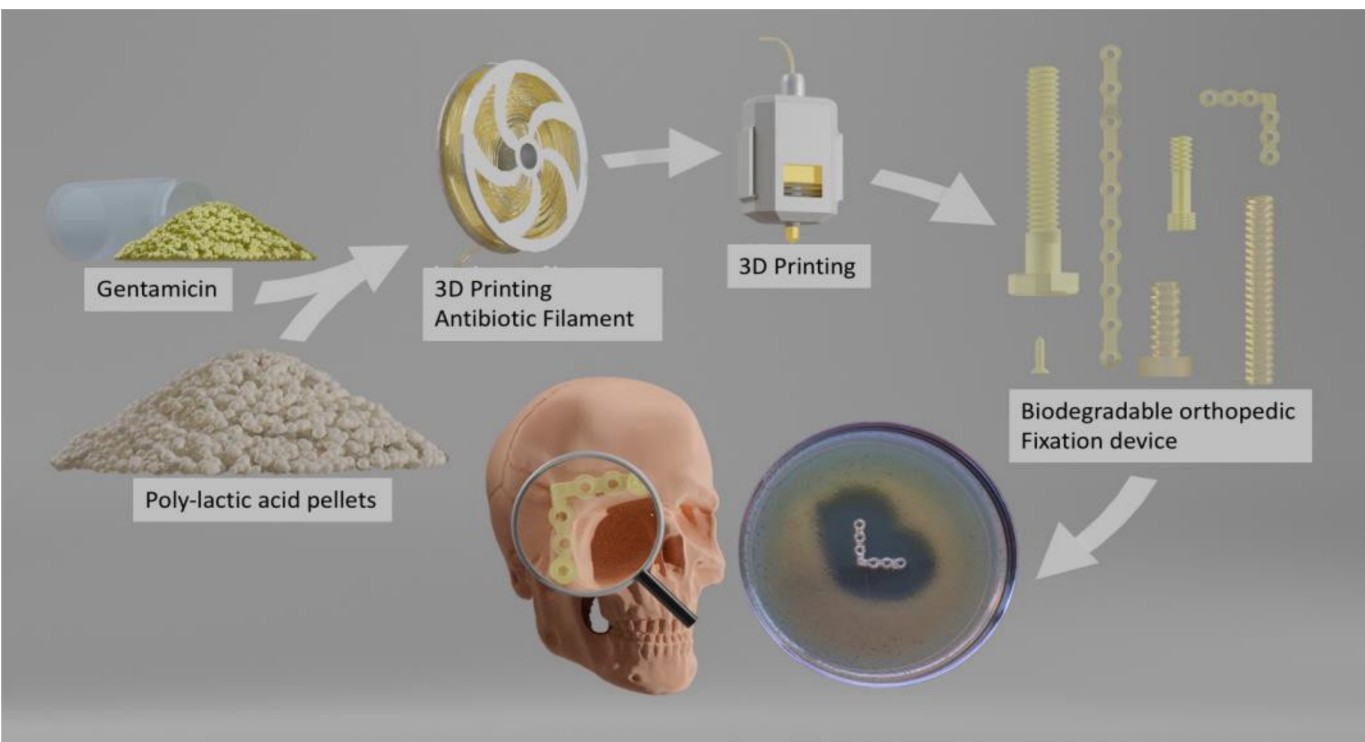

**Figure 6.** Examples of 3D printed devices fabricated from an antimicrobial filament [51].

### 4.3. HNTs as a Drug Delivery System

HNTs have been used as nanocontainers or nanocarriers for drug delivery as well as targeted drug delivery in many research reports [54,55] (Figure 7). Adsorption, intercalation, and tubular entrapment are the most common methods for loading medicines into the lumen or onto the surface of HNT (vacuum method) [56].

HNTs are commonly used as a delivery agent by undergoing some form of alteration, either to the outside surface or to the lumen. This is performed because natural HNTs have a weak affinity with medicines in many circumstances, making prolonged release impractical. Natural HNTs have been reported to be employed as cores for layer-by-layer (LbL) encapsulation in the past. This resulted in greater loading and continuous drug release for up to 100 h, with the creation of tube-end stoppers assisting in drug release prolongation [57,58].

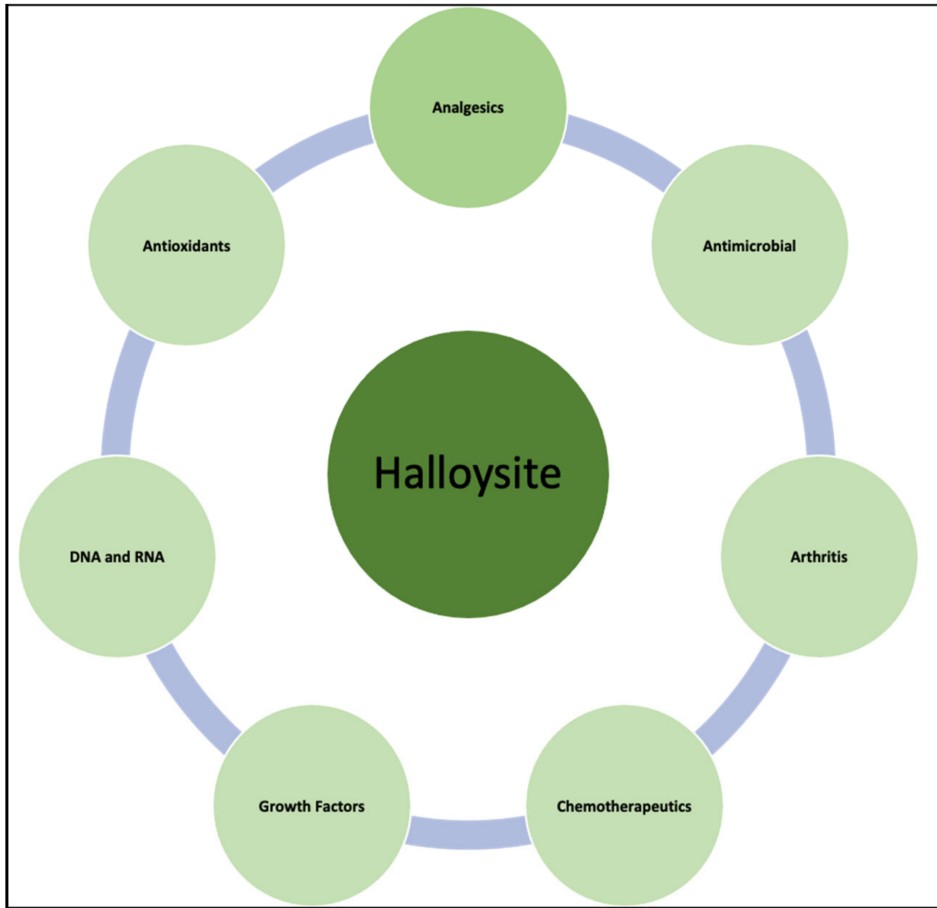

**Figure 7.** Types of drugs loaded onto halloysite nanotubes.

*4.4. Tissue Regeneration*

Reports have shown [59] that surgeons within the USA have been exploring antibiotics-loaded bone cement to mend implants; however, the evolution of metal-based nanoparticles has made many successful alternatives to antibiotics feasible and simple, because nanoparticles not only possess excellent antimicrobial properties, but also possess several other superior properties such as biodegradability, biocompatibility, and higher mechanical strength. Since then, the use of metal nanoparticles has seen increasing demand as an alternative treatment for reducing microbial infections, leading to progress in orthopedic surgery and wound healing, as new suitable biomaterials are being used as bone substitutes because of their structural, biological, and mechanical properties. Furthermore, these metal nanoparticles play a crucial role in tissue engineering due to the properties they contribute to a scaffold. Silver nitrate ($AgNO_3$), copper sulfate ($CuSO_4$), and zinc sulfate ($ZnSO_4$) have all been coated on the HNT external surface for their inherent antimicrobial properties. In this manner, encouraging tissue regeneration while controlling infection may be possible.

Silver (Ag), copper (Cu), and zinc (Zn)-coated HNTs were tested, resulting in the growth restriction of both Gram-negative and -positive bacteria [27–30,52,60–63]. Metal nanoparticles involving Ag have been used to functionalize composite scaffolds that can be used for biomedical applications in which there is no toxic effect, with the scaffolds showing high biocompatibility [62]. It has also been reported that the immobilization of AgHNT and ZnHNTs resulted in superior mechanical strength, antimicrobial properties, and good curing depth with no cytotoxic effects, making them multifunctional composites in dental applications [63].

Metal oxide nanoparticles have been used extensively in antimicrobial applications and wound healing, in particular copper, silver, and zinc (Figure 8) [29,30,64]. In various forms, iron oxide nanoparticles (IONPs) have been used in regenerative medicine applications including tissue engineering and cell therapy [65,66]. Cerium oxide and manganese dioxide polymer composites have been used for the treatment of osteoarthritis, slowing cartilage degradation by inhibiting inflammation and promoting cartilage repair [67]. For these reasons, hydrogels need to be combined with suitable substances or materials that can significantly address these limitations by improving and enhancing their physical, biological, and mechanical properties, which leads to the incorporation of metals with HNT and opens up applications as antimicrobials [64,65] and in drug delivery [64,65], tissue regeneration [66–68], and wound dressing [69]. For example, gold nanoparticles are a good material for bone tissue engineering because they promote osteogenic and adipogenic differentiation from mesenchymal stem cells and influence osteoclast formation from hematopoietic cells while protecting mitochondrial dysfunction in the osteoblastic cell [62]. For an extensive review on a variety of metal oxide biomedical applications see Deo et al., 2019 [69]: limitations of using hydrogel scaffolds in tissue engineering.

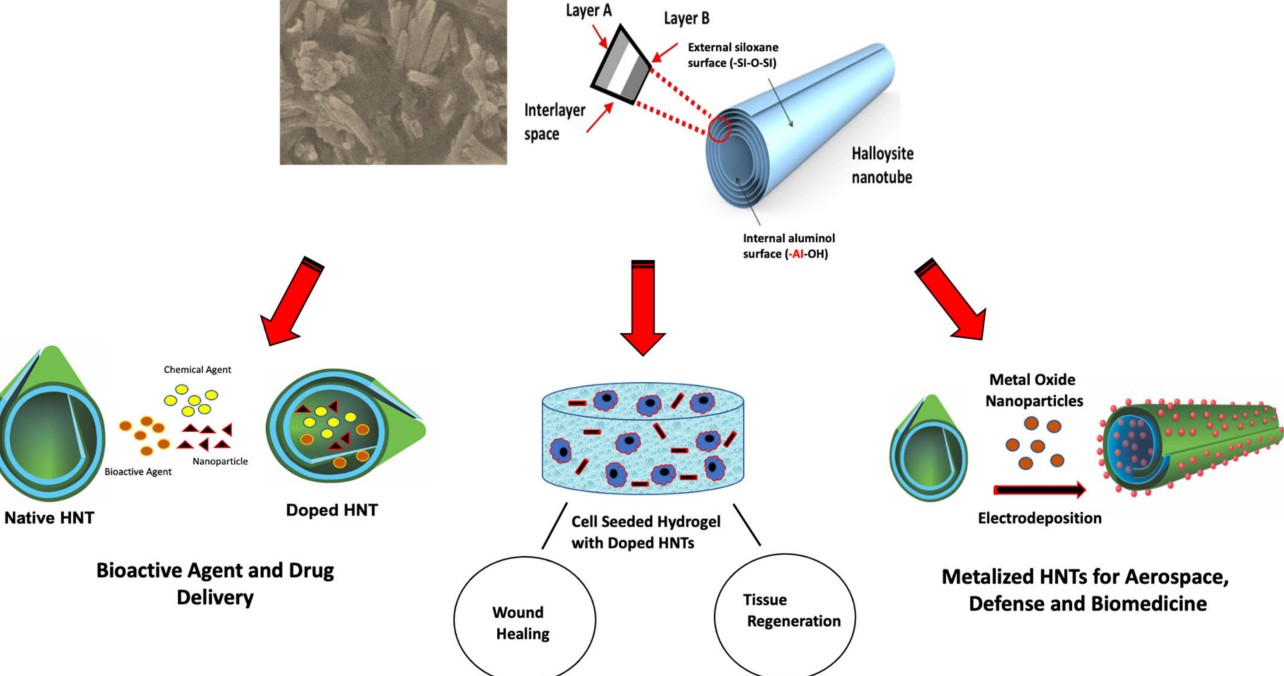

**Figure 8.** Native HNTs have been used as both nanocarriers and nanocontainers for aerospace, biomedicine, defense, and industrial applications.

## 5. Industrial and Military Applications

### 5.1. Radiation Absorptive Composites

Nanomaterials are drawing increased attention as an alternative to lead-based radiation shielding material [70,71]. Lead, while effective, is a poisonous metal, and its effects on the human body are devastating and can lead to persistent vomiting, encephalopathy, lethargy, delirium, convulsions, and coma [72,73]. The type and composition of shielding required depend on the kind of radiation, the activity of the radiation source, the acceptable dose rate, and the environmental conditions, whether industrial, medical, or in space [3,72].

An effective shield results in a significant energy loss at a small penetration distance, minimizing hazardous radiation. Furthermore, suitable shielding material should have a high absorption cross-sectionally for radiation attenuation. Under extreme space conditions, lightweight structures should also have robust mechanical performance, which cannot be achieved by traditional machining, molding, or injection molding [74]. Three-dimensional

printing is a technology that offers a means of fabricating items with precise custom geometries. Three-dimensionally printed constructs can benefit greatly from enhanced flexibility, stretchability, and complex architectures.

Recently, custom 3D printer filaments have been produced from polycaprolactone, polylactic acid, nylon, and polypropylene (Figure 9) [75]. The additives included iron oxide, barium oxide, and gadolinium oxide.

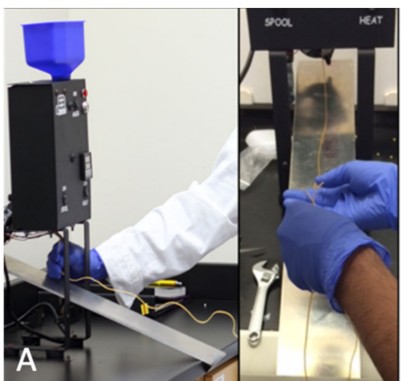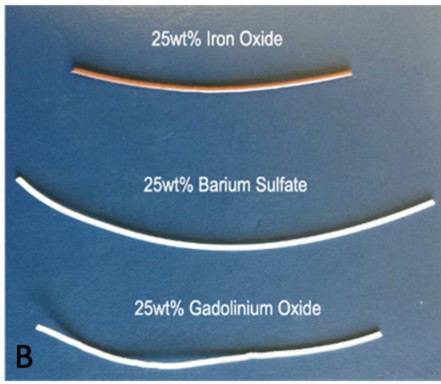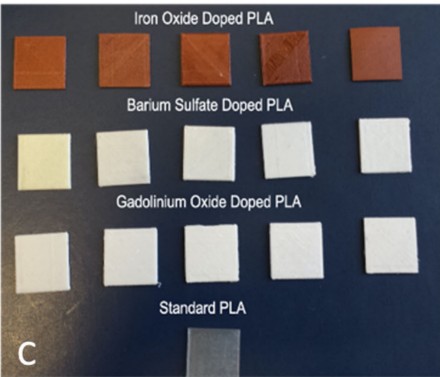

**Figure 9.** (**A**) Customized printer filament being extruded. (**B**) Photographs of iron oxide, barium oxide, and gadolinium oxide printer filaments. (**C**) Three-dimensionally printed PLA doped squares and standard PLA. Reprinted from [76].

*5.2. Electronic Components*

The demand for multifunctional small-size electronic components is increasing day by day. As a result, many researchers are working on fabricating highly conductive and flexible electronic components using conductive filler in various metal and carbon nanocomposite materials [75–77]. Silver nanomaterials are popular because of their high conductivity and thermochemical stability [78,79]. However, the manufacturing cost is relatively high for highly conductive silver nanomaterials. Copper nanomaterial could be a low-cost solution, but its high oxidation properties are a significant hindrance [78]. Accordingly, finding a low-cost, high-performance nanomaterial for electronic components remains a challenge.

HNTs are naturally found, low-cost nanomaterials that can trap highly conductive metal particles in their inner lumen or deposited on their surface (Figure 10). A conductive co-filler of polyaniline–halloysite nanotubes with silver nanoflakes was found to have excellent conductivity and water dispersity, which can be used to fabricate high-performance conductive polymer composites (CPCs) [79,80]. In addition, polyaniline–halloysite nanocomposite shows ultra-low resistivity and excellent electrical stability and can be used to manufacture various capacitive electrodes, electrochemical sensors, and soft electronics [81,82]. Polyvinyl alcohol (PVA) based polydopamine (PDA) coated HNTs with ferric ions create a novel composite material that shows three times more electrical conductivity and almost seven times more energy dissipation than its counterparts. This nanocomposite material can maintain its electrical properties even after 600% strain and 3000% elongation of its original lengths [83]. These properties make the composite ideal for fabricating soft, bendable, wearable electronics, sensors, and catalysts. A clay nanocomposite material can be developed by mixing HNTs with silver-filled epoxy (SFE) using the intercalation method, which is highly feasible for semiconductor and electronics applications. The semiconductor chips achieve desirable conductivity and shear strength at 3%–6% HNT composition [84].

### 5.3. Toxic Waste Removal

The presence of an excessive amount of toxic waste and metal particles in water may cause severe human and animal health issues. The removal of harmful metal particles is highly important for environmental safety. Absorption-based removal methods are the most common and cost-effective techniques to eliminate toxic metal pollutants [85]. Clay minerals are typically an excellent absorbent and have been gaining popularity because they can be found in nature and have unique anti-pollutant properties [86]. Among them, halloysite nanoclay composite materials have emerged as a very efficient absorbent and waste removal agent because of their non-toxic nature, high specific surface area, biocompatibility, and natural presence. Additionally, the absorption capability of HNTs can easily be increased by metalization and surface modification [87].

HNTs are an ideal alternative to remove most of common metal ions from wastewater, such as Cu, lead (Pb), cadmium (Cd), Zn, cobalt (Co), and chromium (Cr) [86–89]. HNTs can naturally remove harmful ingredients from aqueous solution without any modification. However, to increase the load capacity of heavy metal ions, HNTs can be loaded with many types of metal-based composites. One study shows that manganese-oxide-loaded HNT (MHNTs@$MnO_2$) nanocomposite is an excellent sorbent for the removal of lead (II) ions from an aqueous solution, with an absorption capacity up to 59.9 mg/g. Thermodynamic analysis shows that the process is completely spontaneous and endothermic, with no reduction in absorption capacity even after five absorption cycles [87]. Halloysite/$CoFe_2O_4$ magnetic composites can remove tetracycline hydrochloride (TC–HCl) from aqueous solutions [88]. A halloysite-$Fe_3O_4$ composite synthesized via the chemical precipitation method effectively removes metal contaminants from wastewater by magnetic separation. Research shows that a coating of $Fe_3O_4$ on the surface of HNTs significantly increases their absorption affinity toward Cd(II) and Pb(II) contaminants. The highest recorded absorption capacity was 33 and 112 mmol·$kg^{-1}$ for cadmium and lead, respectively, with 10% surface deposition of $Fe_3O_4$ [89]. The 10% and 25% mass ratios are marked as HFe10 and HFe 25, whereas the thermally treated 10% and 25% $Fe_3O_4$ at 400 °C are marked as HFe10K and HFe25K, respectively (Figure 10).

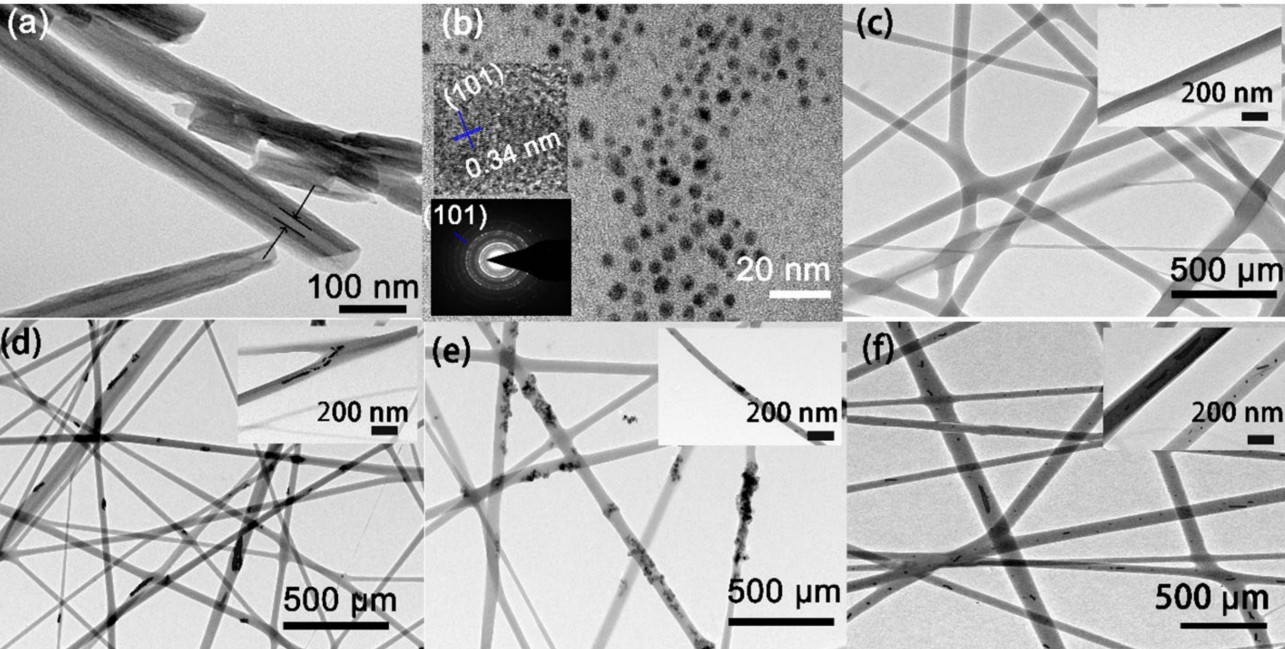

**Figure 10.** TEM images of (**a**) HNTs, (**b**) $Fe_3O_4$ nanoparticles (insets are the selected area electron diffraction pattern and HRTEM image), (**c**) PEO/CS nanofibers, (**d**) $Fe_3O_4$/PEO/CS nanofibers (4% $Fe_3O_4$), (**e**) HNT/PEO/CS nanofibers (4% HNTs), and (**f**) HNT/$Fe_3O_4$/PEO/CS nanofibers (4% $Fe_3O_4$ and 4% HNT). Reprinted with permission from Ref. [90], Elsevier 2018.

Various natural and synthetic industrial dyes are becoming a significant source of toxic material in water. The worldwide production of dye materials is more than $7 \times 10^5$ tons annually. A cost-effective method of removing dyes from wastewater gained much attention in the past few decades [90]. Various metal-based composites of HNTs have been widely used as absorbents to remove different classes of dyes for decades. HNTs modified with iron oxide nanoparticles can effectively remove methylene blue (MB), neutral red (NR), and methyl orange (MO) dyes from an aqueous solution by the magnetic absorption process [91]. HNT/Fe3O4/C hybrid nanocomposite has a heterogeneous structure and shows non-ideal monolayer adsorption behavior. It demonstrates excellent potential for water treatment because of its fast and very good magnetic dye absorption properties [92]. HNTs/TiO$_2$ composite exhibits an enhanced photocatalytic activity in the decomposition of methylene blue. So, many researchers have proposed a combination of TiO$_2$, ZnO, and Ag-loaded HNTs for effectively removing a wide range of dyes based on the dual mechanisms of absorption and photodegradation [93–96].

*5.4. Industrial Catalysts*

Chemical, textile, and pharmaceutical industries use various catalysts to change reaction rates. Metal-based nanomaterials have been used as industrial catalysts for the last few years [97]. Gold (Au) based nanoparticles have emerged as an effective method of chemical catalysis for their unique catalytic properties [98]. The halloysite nanotube (HNT) surface can be modified using gold (Au) particles by a homogeneous deposition–precipitation method to significantly increase the catalytic performance for solvent-free oxidation of benzyl alcohol. The combination of a higher amount of oxidized gold species and the hollow structure of the HNTs make the composite a much more effective and stable catalyst than other nanocomposites [99]. Nano CuO can be loaded with HNTs by the impregnation method to vastly increase the catalytic oxidation performance for the selective oxidation of cyclohexene [100]. Toluene is one of the most-used volatile organic compounds (VOCs) in industry. Cu and Co oxide-loaded HNTs were found to be excellent industrial catalysts for the total oxidation of toluene (Figure 11) [100,101]. The organic synthesis of different alkaline classes is widespread in industrial sectors. Mo salen-supported HNTs (HNTs-Mo-SL) created using a facile chemical surface modification and self-assembly method were found to be a highly reactive catalyst in the epoxidation of a wide range of alkenes, including linear, cyclic, and aromatic alkenes [102].

Halloysite nanotubes are also effective agents for loading photocatalytic materials onto their surfaces and lumens. Photocatalytic HNT nanocomposites remove dyes, toxic chemicals, and drugs from aqueous solutions by photodegradation. Investigation has found that TiO$_2$-HNTs nanocomposite significantly improves photocatalytic activity in the decomposition of methylene blue compared to pristine HNTs [103]. The green synthesis of Ag nanoparticles (AgNPs) loaded on halloysite nanotubes (HNTs) was successfully used to degrade methylene blue. This study found that almost 90% of the pollutant was decomposed photocatalytically within only one hour [104]. Similarly, halloysite nanotube supporting a hybrid CeO$_2$–AgBr nanocomposite shows a much higher photocatalytic activity than pure HNT. The maximum decomposition of methylene orange was 99% within one hour and twenty minutes [105].

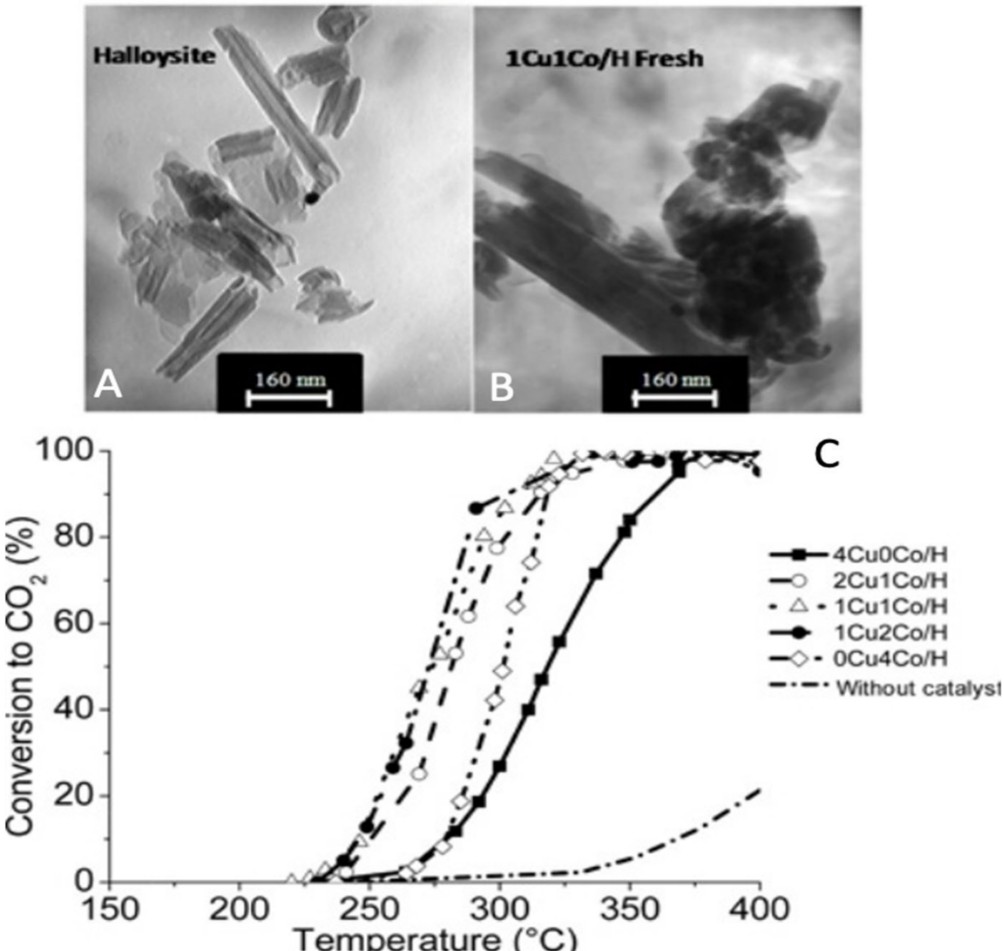

**Figure 11.** Structural stability assessment of the support and the (20 wt%) 1Cu1Co/HNT catalyst. (**A**) Nanotubes observed via TEM before and after catalyst synthesis (**B**). (**C**) XRD evidence after synthesis and a catalytic test. Reprinted with permission from Ref. [102]. Elsevier, 2015.

## 6. Product Market Analysis and Projections

Halloysite is a naturally-occurring nanoparticle. Halloysite deposits are found in China, South Korea, New Zealand, and the US [104]. Halloysite is mined from various sites across the globe. Each form has different features and properties [46]. The cost of unrefined halloysite is estimated to be up to USD 5000/ton, compared to a kaolin–halloysite hybrid, which costs between USD 500 and USD 1000/ton, with pure kaolin going for USD 300/ton [105].

Halloysite is widely used as a bulk additive for polymers such as rubber and in high-tech applications such as hydrogen storage, water purification, carbon capture, removal of toxic compounds from soil, and renewable energy [105]. An assessment of metallized halloysite's environmental impact must be viewed in terms of the application. Every application, type of metal oxide used, fabrication method, polymer type, and additive (including halloysite) will have a different environmental impact.

### 6.1. Metal and Metal Oxide Nanoparticles

Metal oxide nanoparticles are increasingly becoming part of our daily lives, and their use is expected to increase significantly in the next decade. The nano-size and versatility of metal nanoparticles enable them to interact with biomolecules on the surface of and inside cells, leading to enhanced signal transduction and target specificity for diagnostics and therapeutics [106]. These properties enable their use in several biomedical applications, including antimicrobial therapy, thermal ablation, radiotherapy enhancement,

highly sensitive diagnostic assays, tissue repair, and drug and gene delivery [107,108]. Current research efforts have enhanced our understanding of metallic nanoparticles to reduce the toxicity issues associated with their usage. Accordingly, new developments are expected to accelerate the production and sale of such materials. According to current clinical lab standards, our mHNTs have been demonstrated in extensive testing to be cytocompatible.

The market is expected to witness steady growth driven by increasing application areas of metal nanoparticles resulting from rapid industrialization and urbanization. Increasing research and development activities are guided by market need for enhanced product applications. North America is currently the largest and most-developed market for metal and metal oxide nanoparticles, with an estimated market value of 25.26 billion by 2022 [109]. This is mainly due to significant venture capital investment motivated by high-demand biomedical (strontium: osteoporosis), cosmetic (zinc: sunscreen), and pharmaceutical (copper, gold, iron: anti-cancer) applications. Europe, Asia Pacific, and Japan are seeing significant growth, expected to increase as demand increases and this market matures [110].

Four types of nanoparticles, including copper, gold, and silver, accounted for more than 90% of the market in 2018 [50,52]. This sector looks set to increasingly use copper and silver in clinical trials using nanoparticles to deliver anticancer drugs, antimicrobial implants and packaging, industrial coatings, and biomedical applications.

### 6.2. Protective Films and Coatings

Custom films and coatings can be custom designed to achieve specific application requirements. Three-dimensional printing, electrospray, and blow spinning enable the production of customized coatings and surfaces with optimal size, shape, mechanical properties, and functionalities. This coating technology can be applied to exterior vehicle surfaces and high-touch surfaces (grab handles, doorknobs, and floor coverings) with capabilities comparable to state-of-the-art vehicle paints and topcoats. The application of antimicrobial/viral surface coatings that are application-specific and provide sustained antimicrobial protection despite wear and environmental exposure with sustained self-decontamination capability is under intense research.

Protective films are produced using various polymers including polyvinyl chloride (PVC), low-density polyethylene, polyethylene–cellophane, thermoplastic polyurethane, and others based on the construction material [111]. A protective film or surface coating helps prevent pathogenic material from entering medical devices and leading to infections [112]. Polyurethanes have also been studied due to their high mechanical strength. Protective films composed of polyurethane offer strong resistance against water, fungus, and abrasion [113]. Traditionally, the healthcare industry uses glass and plastic packaging components to protect medical devices from the outside environment. Currently, the industry relies on high-barrier plastic films composed of single polymers, laminations, metallized films, and various specialized coatings for better product protection. During the last ten years, paper was the most commonly used material for medical device packaging, but it has been rapidly replaced in recent years by plastic and silicone films. Medical device protection films can be segmented by type into single films, laminations, and co-extrusions films.

No matter the application, the addition of MNPs is used to enhance material properties (thermal resistance/conductance, elasticity, tensile and compression resistance, abrasion and wear) [114]. Additionally, metal nanoparticles impart other functionalities, such as anti-fouling and anti-microbial properties, elasticity, friction resistance, and protective films for surgical and medical instruments, ophthalmic devices, dental devices, and others [115].

Some of the significant drivers of protective films for the medical device market include the healthcare industry due to stricter regulatory standards and increased adoption of protective film and coating technology [12]. Globally, governments are spending more money to provide better healthcare services. For instance, the U.S. has the highest per capita healthcare spending globally. NHE grew 2.7% to $4.3 trillion in 2021, or $12,914 per

person, and accounted for 18.3% of Gross Domestic Product (GDP) [12]. Overall, healthcare spending worldwide is growing at a rapid rate.

Healthcare spending is also rising among developed countries such as France, Germany, and Japan, and developing countries such as India and China are focused on providing better healthcare services to their large population bases. This pattern is expected to drive the demand for protective films and coatings.

*6.3. Textiles*

This market is projected to witness steady growth in the next few years because of the increasing application of metal nanoparticles resulting from rapid industrialization and urbanization. Nanotechnology has stimulated novel, innovative textile products termed "nanotextiles." Known applications of nanotextiles are water or stain repellents and antibacterial textiles containing nanoparticles such as quartz ($SiO_2$) or silver (Ag) [116,117]. However, the number of nanotextiles on the market is limited, and market penetration has been slight. To use nanotechnology, the textile industry must deal with new technologies requiring investments, safety issues, and regulations [118]. The unique requirements of current manufacturing methods are costly, inefficient, and require the use of harsh compounds requiring remediation after production [119]. A more straightforward and less wasteful process for producing nanotextile functional coatings incorporating novel and new attributes/capabilities is highly desirable.

## 7. Future Work and Impact of mHNTs

Excellent physio-chemical properties and growing usage of nanomaterials in the electronics, healthcare, aerospace, and textiles industries are expected to drive the market forward. The added functionalities of metallized HNTs will foster accelerated growth. As halloysite is mined and generally inexpensive, its adoption not only as bulk filler but also as a nanocontainer is expected to grow significantly.

The fabrication of halloysite-nanotube-supported metal nanoparticles has many potential uses across industries. In biomedicine, the inherent biological properties offer much prospect in applications such as antimicrobial surfaces, [29,30,60] sustained and target drug delivery, [51,56,64] tissue engineering [6,111], and theragnostic systems [112]. Industrial applications include use as an absorbent and detoxifying agent, use in radiation and EMP protection, and energy applications.

Many concerns regarding nanoparticle safety exposure to a larger population, which is expected to grow significantly, remain. Potentially harmful effects are expected to trigger the implementation of more stringent regulations in terms of manufacture and disposal. This outcome will be a challenge to industry growth. In contrast, halloysite is noncytotoxic and biocompatible in physiologically relevant amounts. The challenge is how materials with halloysite added are recycled or remediated for low environmental impact. A natural adsorption capability that shows excellent ability will play a role in environmental clean-up and remediation.

**Author Contributions:** Conceptualization, all authors; validation, D.K.M.; investigation, all authors; resources, D.K.M.; writing—original draft preparation, D.K.M., P.A.S.M.; writing—review and editing, A.-R.M., F.A., M.J.P.B., D.K.M.; supervision, D.K.M.; project administration, D.K.M.; funding acquisition, D.K.M. All authors have read and agreed to the published version of the manuscript.

**Funding:** This manuscript was funded by NASA EPSCoR Rapid Response Research (R3) program award 22-2022 R30015.

**Institutional Review Board Statement:** Not applicable.

**Informed Consent Statement:** Not applicable.

**Data Availability Statement:** Not applicable.

**Conflicts of Interest:** The authors declare no conflict of interest.

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
