# Peer review of "Metallizing the Surface of Halloysite Nanotubes—A Review"

_coatings, doi:10.3390/coatings13030542_

Round 1
Reviewer 1 Report
This work is well-organized. I only have four comments to improve the manuscript's quality.
#1 Please make a nomenclature list or give a full name for the abbreviation (e.g., SEM) first appearing in the main text.
#2 Line 97, the labels (A, B, C, and D) seem to be wrong.
#3 Table 3, please clarify the definitions of adsorption and absorption. In environmental applications, absorption is often used to describe mass transportation from gas to liquid. I don’t think HNT composites can remove dyes via absorption. In fact, adsorption is the major mechanism to remove pollutants from the liquid phase using carbonaceous materials.
#4 I highly recommend the authors add a section discussing the regeneration, reuse, or disposal of used mHNTs. Also, add a part to Section 6 to discuss (or give perspectives) the life cycle assessment, production cost evaluation, and carbon emission of mHNTs’ production/application.
Author Response
Reviewer #1
The authors thank the reviewer for their time and effort in improving our manuscript. Additionally, the corresponding author offers his apology for the draft you reviewed. It was not the final submission. He was working with the editor to alter our submission.
This work is well-organized. I only have four comments to improve the manuscript's quality.
Author’s Response: We that the reviewer for the positive comments. We have addressed your comments below and in our revised manuscript.
1). Please make a nomenclature list or give a full name for the abbreviation (e.g., SEM) first appearing in the main text.
Author’s Response: We have provides abbreviations as needed.
2). Line 97, the labels (A, B, C, and D) seem to be wrong.
Author’s Response: This has been corrected.
3). Table 3, please clarify the definitions of adsorption and absorption. In environmental applications, absorption is often used to describe mass transportation from gas to liquid. I don’t think HNT composites can remove dyes via absorption. In fact, adsorption is the major mechanism to remove pollutants from the liquid phase using carbonaceous materials.
Author’s Response: All tables were removed in our final draft.
#4 I highly recommend the authors add a section discussing the regeneration, reuse, or disposal of used mHNTs. Also, add a part to Section 6 to discuss (or give perspectives) the life cycle assessment, production cost evaluation, and carbon emission of mHNTs’ production/application.
Author’s Response: We have added a section on these areas.
Reviewer 2 Report
Whole the manuscript should be completely checked by native editor.
All of the references should be write in same style:
For example in ref. 1,2,3 and so on the dates were written in parenthesis but, ref 8, 10 and so on were witten in other style!
The section “Tissue Regeneration” should be discussed in more detailed.
Data in Table. 1 and 3 should be repeated at least 3 times and will be reported in (Average+_Standard deviation) and then P-Value should be calculated.
The section “future prospective” should be added at the end of the manuscript.
Author Response
Reviewer #2
The authors thank the reviewer for their time and effort in improving our manuscript. Additionally, the corresponding author offers his apology for the draft you reviewed. It was not the final submission. He was working with the editor to alter our submission and the draft was sent out for review.
1). The whole manuscript should be completely checked by native editor.
Author’s Response: We have had the manuscript reviewed by an English speaker.
2). All of the references should be write in same style:
For example, in ref. 1,2,3 and so on the dates were written in parenthesis but, ref 8, 10 and so on were written in other style!
Author’s Response: We have revised our references to ensure a consistent format.
2). The section “Tissue Regeneration” should be discussed in more detailed.
Author’s Response: We have added a section on this area.
3). Data in Table. 1 and 3 should be repeated at least 3 times and will be reported in (Average+_Standard deviation) and then P-Value should be calculated.
Author’s Response: All tables were removed in our final draft.
The section “future prospective” should be added at the end of the manuscript.
Author’s Response: We have additional text in section 7. Future Work and Impact of mHNTs
Reviewer 3 Report
1. The aspect ratio of the figures needs to be adjusted.
2. The font size of the figures needs to be adjusted.
Author Response
Reviewer #3
1). The aspect ratio of the figures needs to be adjusted.
Author’s Response: The aspect ratio of the figures is determined by the publisher. We can request adjusting the aspect ratio if the paper is accepted and during the review of the proof.
- The font size of the figures needs to be adjusted.
Author’s Response: We can request adjusting font size if the paper is accepted and during review of the proof.
Reviewer 4 Report
The authors present a review on the techniques of metallization of HNT surfaces and their applications. First, the review deals out with a brief overview of the current methods for metalizing the HNT surface. Hence, the authors provide a detailed study on specific applications of metal-coated HNTs and in combination with polymer composites in anti-infective therapy, immunotherapy, dentistry, regenerative medicine, and wound healing. In turn, the commercialization potential of metal-coated HNTs is also discussed.
The review is well written and well organized. The references are appropiate and the overview as given by the authors could help the readers that first are approaching HNTs.
I just point out the figure 10 has been forgotten or deleted, there being only the caption.
Many figures are not original and they should be accompanied by the original refeences with the correspondent permission.
Analogously, Table 2 and Table 3 are mentioned but not included to the text.
After such mandatory changes the review will deserve the publication.
Author Response
Reviewer #4
The authors present a review on the techniques of metallization of HNT surfaces and their applications. First, the review deals out with a brief overview of the current methods for metalizing the HNT surface. Hence, the authors provide a detailed study on specific applications of metal-coated HNTs and in combination with polymer composites in anti-infective therapy, immunotherapy, dentistry, regenerative medicine, and wound healing. In turn, the commercialization potential of metal-coated HNTs is also discussed.
The review is well written and well organized. The references are appropriate, and the overview as given by the authors could help the readers that first are approaching HNTs.
Author’s Response: The authors thank the reviewer for their time and effort in improving our manuscript. Additionally, the corresponding author offers his apology for the draft you reviewed. It was not the final submission. He was working with the editor to alter our submission and was not aware that it was sent out for review.
I just point out the figure 10 has been forgotten or deleted, there being only the caption.
Author’s Response: Figure 10 has been included in our revised manuscript.
Many figures are not original, and they should be accompanied by the original references with the correspondent permission.
Author’s Response: Figures 1 through 5, and 7 through 9 are original. Figure 6 was a figure made by the publisher when it was featured in reference 51. It has not been published and figures 10 add 11 are reproduced with permission of the publisher.
Analogously, Table 2 and Table 3 are mentioned but not included to the text.
Author’s Response: All tables were removed in the final submitted manuscript.
Reviewer 5 Report
Overall, the authors draw a detailed summary of HNTs, surface modifications of HNTs, and its related applications in biomedical, industrial, and military fields. Sufficient amount of literature review is included with acceptable level of discussion and summarization. However, the format, the figures, consistency, and the details really need more efforts. A major revision must be conducted before re-considering of acceptance.
1. A formal introduction part is necessarily required. In this manuscript, the authors simply separate the manuscript as 7 parts without a formal introduction part illustrating their own view toward HNT and metallization of its surface and the following applications.
2. It is really weird to find out the Figure 2 is first discussed, rather than Figure 1; while Figure 1 occurs in section 4 for the first time. It is quite obvious that the authors pay little attention to the details. It is strongly recommended that the authors should carefully check the details again.
3. Similarly, there is no Table in this manuscript. Could the authors explain why “table 2” is mentioned for multiple times? If more of these details are found, I would seriously doubt whether there exists plagiarism.
4. As a review paper, the figures need to be detailly taken care of. For example, Fig.1A, we can only see “100 nm” and we can’t even see the scale bar. And a series of figures would be much better if the authors can conduct a reasonable recreation.
Author Response
Reviewer #5
Overall, the authors draw a detailed summary of HNTs, surface modifications of HNTs, and its related applications in biomedical, industrial, and military fields. Sufficient amount of literature review is included with acceptable level of discussion and summarization. However, the format, the figures, consistency, and the details really need more efforts. A major revision must be conducted before re-considering of acceptance.
Author’s Response: The authors thank the reviewer for their time and effort in improving our manuscript. Additionally, the corresponding author offers his apology for the draft you reviewed. It was not the final submission. He was working with the editor to alter our submission and was not aware that it was sent out for review.
1). A formal introduction part is necessarily required. In this manuscript, the authors simply separate the manuscript as 7 parts without a formal introduction part illustrating their own view toward HNT and metallization of its surface and the following applications.
Author’s Response: A formal introduction has been included and subdivide into sections on structure, surface modification, and cyto- and biocompatibility.
2). It is really weird to find out the Figure 2 is first discussed, rather than Figure 1; while Figure 1 occurs in section 4 for the first time. It is quite obvious that the authors pay little attention to the details. It is strongly recommended that the authors should carefully check the details again.
Author’s Response: We have corrected this in our revision.
3). Similarly, there is no Table in this manuscript. Could the authors explain why “table 2” is mentioned for multiple times? If more of these details are found, I would seriously doubt whether there exists plagiarism.
Author’s Response: Tables were not included in our final submission. Plagiarism was not involved.
4). As a review paper, the figures need to be detailly taken care of. For example, in Fig.1A, we can only see “100 nm,” and we cannot even see the scale bar. Moreover, a series of figures would be much better if the authors could conduct a good recreation.
Author’s Response: We have added scale bars to figure 1. The authors regretfully do not understand the instructions suggested in the last sentence above.
Round 2
Reviewer 2 Report
The revised manuscript is acceptable.
Reviewer 5 Report
The revised manuscript looks acceptable to me. I have no further comments.